# Late Paleozoic Tectonic Evolution of the Qinling Orogenic Belt: Constraints of Detrital Zircon U-Pb Ages from the Southern Margin of North China Block

**Wentao Yang [1],\*, Te Fang [1], Yanpeng Wang [2]**  **and Hao Sha [1]**

[1] School of Resources and Environment, Henan Polytechnic University, Jiaozuo 454000, China; edufte@163.com (T.F.); shahao2001@126.com (H.S.)
[2] School of Design and Art, Henan University of Technology, Zhengzhou 450001, China; wangyp2017@126.com
\* Correspondence: yangwt@hpu.edu.cn

**Abstract:** The tectonic evolution of the Qinling Orogenic Belt in the Late Paleozoic has long been controversial, especially due to the limitation of the Mianlue Ocean subduction time. Basin formation and sedimentary development in the southern North China Block are closely related to the tectonic evolution of the Qinling Orogenic Belt, which is an effective entry point to study basin–mountain interaction. We present new detrital zircon U–Pb data from the Shihezi Formation in the Luonan area in the southern margin of the North China Block. The results show that the bottom sample has two major peaks at 288 Ma and 448 Ma, with weak peaks at 908, 1912 Ma and 2420 Ma. The top sample has one major peak at 297 Ma, with weak peaks at 1933 Ma and 2522 Ma. Combined with the published paleocurrent data and lithofacies paleogeography, the sediments of the bottom sample were sourced from the North Qinling Belt, Inner Mongolia Palaeo-Uplift and the basement of the North China Block. The top sample originated mainly from the Inner Mongolia Palaeo-Uplift and the basement of the North China Block. Comparing the obtained zircon U-Pb ages with the published relevant data in the North China Block, it is found that the provenance area shifted from the Qinling Orogenic Belt to the Inner Mongolia Paleo-Uplift in the Late Carboniferous–Permian, and the Qinling Orogenic Belt could hardly provide provenance for the southern North China Block in the Middle Permian. The uplift of the Qinling Orogenic Belt in the Late Carboniferous may be the continuation of Caledonian orogeny in the Early Paleozoic, whereas the uplift of the Inner Mongolia Palaeo-Uplift is related to the tectonic evolution of the Central Asian Orogenic Belt during the Late Paleozoic. This tectonic transformation occurred when the Qinling Orogenic Belt no longer supplied sediments to the southern North China Block in the Middle Permian, and the Mianlue Ocean subduction did not occur until at least the Late Permian.

**Keywords:** Late Palaeozoic; Mianlue Ocean; sedimentary response; southern North China Block; zircon U-Pb dating

## 1. Introduction

The Qinling Orogenic Belt is located in the central part of China, which is of great significance to the formation of the Chinese mainland and is also an important area for understanding the evolution of the Paleo-Tethys Ocean. Previous studies generally suggested that the Qinling Orogenic Belt experienced a long-term, multi-stage tectonic evolution process, and finally, the South China Block, the Qinling microblock and the North China Block converged and collided with each other along the Mianlue suture in the Triassic [1,2]. The way of assembly was an oblique collision from east to west. The eastern collision occurred at the end of the Permian, while the western collision may have reached the Late Triassic [3–5].

Triassic orogenesis in the Qinling Orogenic Belt is not only supported by the evidence of magmatic petrology [6], but also by metamorphic petrology [7–9] and sedimentary

petrology [10–12]. However, there is no consensus on when the Mianlue Ocean began to subduct after the Silurian rifting. Although the anorthosite and diabase from the Sanchazi area and andesite at the Nanping area, respectively, yielded U-Pb zircon ages of 300 ± 61 Ma [8], 295~264 Ma [13] and 246 ± 3 Ma [14], indicating that the Mianlue Ocean subducted from ca. 300 Ma to 250 Ma, these ages have a wide margin of error. On the other hand, if an East-Pacific-type continental margin existed in the Qinling Orogenic Belt [15], it is not consistent with the extensive Permian sedimentary strata in the southern North China Block and the Qinling area. The major constraint is the lack of Late Paleozoic arc volcanism in the Qinling Orogenic Belt.

Orogenic belts not only control the basin evolution but also supply a large amount of sediments to them. These sediments, in turn, record important information about the tectonic evolution of the orogenic belt. Therefore, the provenance analysis of sediments can be effectively applied to the study of basin–mountain interaction, providing important constraints for restoring the paleogeographical features and tectonic evolution of the basin–mountain system. To date, most of the studies speculate on the late Paleozoic tectonic evolution process of the Qinling Orogenic Belt from the sedimentary strata on the North China Block [16–20]. However, these research areas are mainly concentrated in the inner North China Block, which is far away from the Qinling Orogenic Belt, making a slow or delayed response to the orogenesis. This paper focuses on the Middle Permian Shihezi Formation in the Luonan area of the southern margin of the North China Block, using the method of sedimentary petrology analysis and detrital zircon analysis to investigate provenance information. These sediments have a particular location very proximal to the Qinling Orogenic Belt, allowing us to reconstruct the Paleozoic tectonic evolution of this orogenic belt with heightened sensitivity.

## 2. Geological Setting

### 2.1. Units in the Qinling Orogenic Belt

The northern boundary of the Qinling Orogenic Belt is the Lingbao–Lushan–Wuyang fault adjacent to the North China Block, and the southern boundary is connected with the South China Block by the Mianlue–Bashan–Xiangguang fault. The orogenic belt can be divided into four tectonic units from north to south, including the southern margin of the North China Block, the North Qinling Belt, the South Qinling Belt and the northern margin of the South China Block. They are divided by the Luonan–Luanchuan fault, the Shangdan Suture Zone and the Mianlue Suture Zone, respectively (Figure 1b).

The southern margin of the North China Block was drawn into the orogenesis in the Triassic. The crustal composition of the southern margin of the North China Block has an obvious double-layer structure, an Archean–Paleoproterozoic crystalline basement and a Meso-Neoproterozoic caprock series [2]. The basement rocks are mainly composed of Taihua Group gneiss series and Dengfeng Group medium- and high-grade metamorphic rocks. The caprock is mainly composed of Xionger Group volcanic rock, Guandaokou Group carbonate rocks, Luanchuan Group low-metamorphic clastic rock–carbonate rock association, Luonan Group dolomite, Ruyang Group low-metamorphic clastic rock–carbonate rock association and Taowan Group metamorphic rock as well as sporadically exposed Cambrian, Permian, Cretaceous and large-area Cenozoic sedimentary strata, related to stable sedimentation and weaker magmatic activity [15].

The North Qinling Belt, from north to south, can be divided into the Kuanping Group, Erlangping Group, Qinling Group, Songshugou Ophiolite and Danfeng Group, which are separated by large shear zones or fault zones. The Kuanping Group comprises Meso-Neoproterozoic metabasites and Neoproterozoic metasedimentary rocks, which might have formed in the Kuanping Ocean [21,22]. The Paleozoic Erlangping Group is a set of metavolcanic and metasedimentary rock assemblages dominated by low and medium metamorphism. It is mainly composed of ophiolite units, clastic sedimentary rock and carbonate rock, and the metamorphic degree is from low greenschist facies to amphibolite facies [23]. The Paleoproterozoic Qinling Group is the oldest unit in the North Qinling

Belt. It is mainly a set of medium and deep metamorphic complex series. The main body is characterized by the development of various gneiss, amphibolites and marbles, whose protoliths are clastic rocks, limestones and interlayers of continental tholeiitic lavas, respectively. It is intruded by Neoproterozoic, Paleozoic and Cretaceous granites and contains Early Paleozoic high-pressure/ultra-high-pressure metamorphic rocks [24]. The Proterozoic Songshugou Ophiolite is mainly composed of metamorphic peridotite and tholeiite, which is the largest ultramafic complexes exposed in the North Qinling Belt [25]. The Early Paleozoic Danfeng Group is mainly volcanic sedimentary rocks, the metamorphic degree is greenschist facies to low amphibolite facies, and the volcanic eruption sedimentary sequence can be seen locally. According to isotope geochemical dating, the ophiolite was formed at 534~457 Ma. It is considered to be the product of the Early Paleozoic oceanic crust subduction and ocean basin closure [2].

The South Qinling Belt shows some similarities to the northern margin of the South China Block [1,26]. It is generally believed that the South Qinling Belt includes a Neoarchean–Paleoproterozoic crystalline basement and Meso-Neoproterozoic metamorphic rocks. Sinian–Middle Triassic marine sedimentary rocks and Mesozoic–Cenozoic continental sedimentary rocks developed on these basements [2]. In addition, Early Mesozoic granitic magmatism is also widely developed in the South Qingling Belt [27].

The northern margin of the South China Block is mainly composed of an Archean–Early Proterozoic crystalline basement and a Meso-Neoproterozoic metavolcanic–sedimentary rock series, as well as Paleozoic and Mesozoic–Cenozoic sedimentary rocks [28]. The Archean–Early Proterozoic basement mainly includes Kongling complexes and Houhe complexes, which are composed of TTG felsic migmatitic gneiss, plagioclase amphibolite and metasedimentary rock series. The Meso-Neoproterozoic basement includes the Shennongjia Group in the Huangling area, the Xixiang Group in the Hannan–Micangshan area and the Bikou Group in the Bikou Terrane [2]. These basement complexes were all intruded by Neoproterozoic mafic and granitic magmas [1]. Sinian and Paleozoic neritic sedimentary rocks rest unconformably on the underlying basement rocks, including Sinian clastic rock–limestones, Cambrian Ordovician carbonate rocks, Silurian siltstones and Permian–Middle Triassic carbonate rocks.

### 2.2. Paleogeography and Stratigraphy

The basement of the North China Block is generally considered to have been finally formed by the collision and amalgamation of the eastern and western blocks along the Central Orogenic Belt at about 1.85 Ga [29,30], which is considered to be the aggregation of the Columbia supercontinent [31,32]. Subsequently, a series of rift basins developed in the North China Block at about 1800 Ma, mainly including the Zhaertai Bayan Obo Huade rift belt in the northwest margin, the Yanliao rift belt in the north central margin and the Xionger rift belt in the southern margin [33,34]. Then, Mesoproterozoic–Cambrian sedimentary caprock overlaid on the rift sediments [35–37]. After the Middle Ordovician, the North China Block was subducted by the Paleo-Asian Ocean in the north and the Proto-Tethys Ocean in the south, causing the uplift and denudation of the North China Block, resulting in the absence of strata from the Late Ordovician to the Early Carboniferous [38,39]. In the Late Carboniferous, the North China Block subsided and received sedimentation again, and thus the Late Paleozoic filling evolution history began. Transgression initially extended from the northeast during the Late Carboniferous to the Early Permian, but regression occurred in the southeast in the Late Permian [40]. The sedimentary evolution of the Upper Paleozoic mainly experienced three stages: marine, transitional and continental depositional systems [41,42].

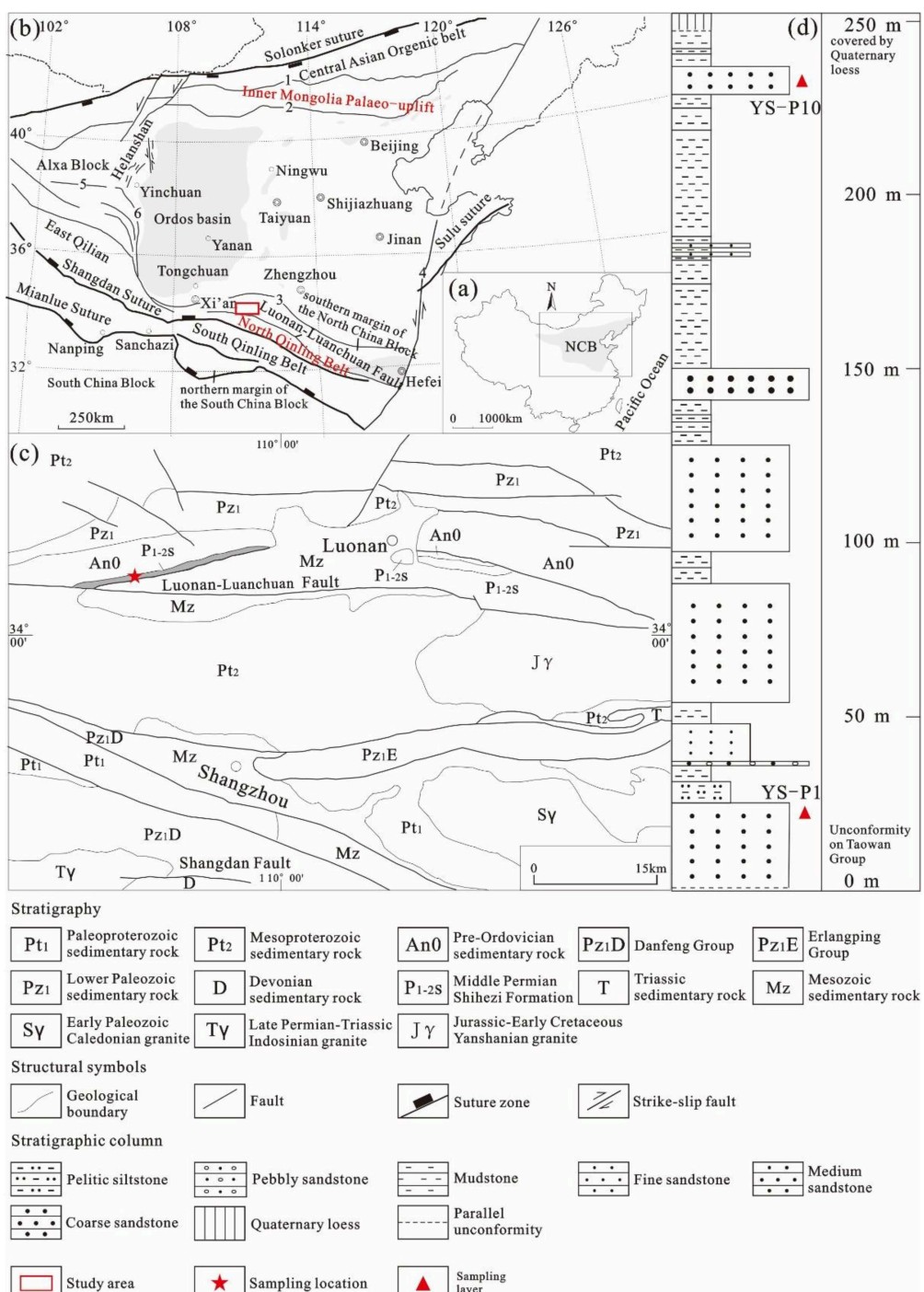

**Figure 1.** (**a**) Inset showing the location of North China Block (NCB) in China; (**b**) tectonic map of North China Block and its adjacent areas (modified from Liu [43]); 1 Chifeng–Bayan Obo fault; 2 Jining–Longhua fault; 3 Lingbao–Lushan–Wuyang fault; 4 Tanlu fault; 5 Longshoushan fault; 6 Baoji–Guyuan fault; (**c**) geological map of Luonan (modified from Chen [44]); gray area represents the study area; (**d**) stratigraphic column of Shihezi Formation.

Extensive deltaic depositional systems developed southward in the North China Block during the Permian and might have reached the southern boundary of the basin or even further south [44,45]. The Permian strata in Miaowan, Luonan area is located in the north of the Luonan–Luanchuan fault (Figure 1c) and is uncomformably on the Early Paleozoic Taowan Group (Figure 2a), with a thickness of more than 246 m. The lower part is mainly gray-white, thick-layered, medium–coarse-grained sandstones and pebbly sandstones

interbedded with gray-yellow and gray-green mudstones (Figure 2b). Parallel beddings (Figure 2c) and cross beddings (Figure 2d–e) are developed in the sandstones. The upper part is mainly grayish green, grayish yellow and purplish red mudstones interbedded with grayish yellow, thick-layered, fine-grained sandstones (Figure 2f), with several thick coal seams (Figure 2i). The top is denuded and covered by Quaternary loess. The Permian strata in this area is a coal-bearing sedimentary strata of fluvial–delta facies. However, the strata are relatively thin and lack marker beds, so it is difficult to further divide them. Previous studies generally considered them as the Shihezi Formation [44].

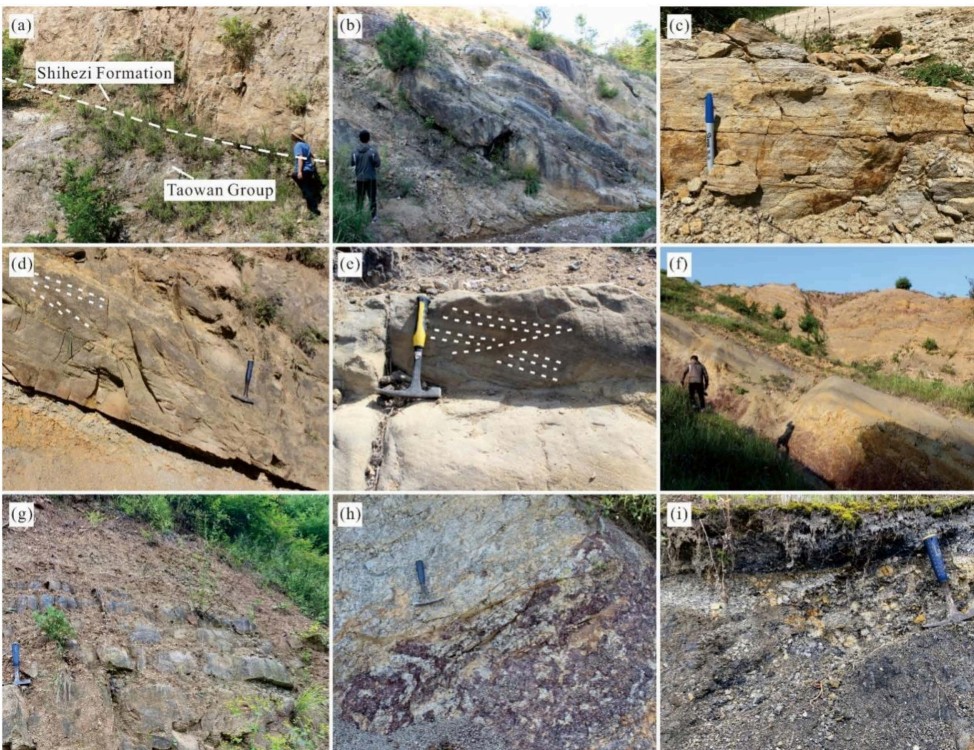

**Figure 2.** Field stratigraphic section and sedimentary structure of Shihezi Formation in Luonan area; (**a**) Shihezi Formation is unconformable above Taowan Group; (**b**) yellow, thick-layered sandstones; (**c**) parallel bedding sandstones; (**d**) tabular cross bedding sandstones; (**e**) wedge cross bedding sandstones; (**f**) profile of upper Permian strata; (**g**) thin mudstones intercalated with thin sandstones; (**h**) purple mottled mudstone; (**i**) coal-bearing strata.

## 3. Samples and Analytical Methods

In this study, two samples of siliciclastic rocks were collected from the bottom and top of the Shihezi Formation, respectively. YS-P1 was collected from the gray-white, medium-grained quartz sandstone at the bottom, and YS-P10 was collected from the earthy-yellow, medium–fine-grained quartz sandstone at the top (Figure 2).

U-Pb dating and trace element analysis of zircon were simultaneously conducted via LA-lCP-MS at the Wuhan Sample Solution Analytical Technology Co., Ltd., Wuhan, China. Detailed operating conditions for the laser ablation system and the ICP-MS instrument and data reduction are the same as those in the description by [46]. Laser sampling was performed using a GeolasPro laser ablation system that consists of a COMPexPro 102 ArF excimer laser (wavelength of 193 nm and maximum energy of 200 mJ) and a MicroLas optical system. An Agilent 7700e ICP-MS was used to acquire ion-signal intensities. Helium was applied as a carrier gas. Argon was used as the make-up gas and mixed with the carrier gas via a T-connector before entering the ICP. A wire signal smoothing device is included in this laser ablation system [47]. The spot size and frequency of the laser were set to 32 μm and 5 Hz, respectively. Zircon 91,500 was used as the primary reference

material [48] for U-Pb dating. Glass NlST610 and GJ-1 were used as the first and second reference materials, respectively, for trace element calibration. Each analysis incorporated a background acquisition of approximately 20–30 s followed by 50 s of data acquisition from the sample. Excel-based software, ICPMSDataCal, was used to perform the offline selection and integration of background and analyzed signals, time-drift correction and quantitative calibration for trace element analysis and U-Pb dating [49]. Concordia diagrams and weighted mean calculations were made using Isoplot/Ex_ver3.

## 4. Result

### 4.1. Zircon Cathodoluminescence Image

It can be seen from the zircon cathodoluminescence (CL) images that most zircon grains are euhedral and fragmented. The grains lengths range from 50 and 200 μm. There is a significant variation in the internal structure and zonation of the detrital zircons. The internal structure can be divided into two types: the first type of zircon has oscillatory zoning and is euhedral, which is a typical magmatic zircon. The second type of zircon shows homogenous internal structures with dim colors or a weakly zonal structure (Figure 3).

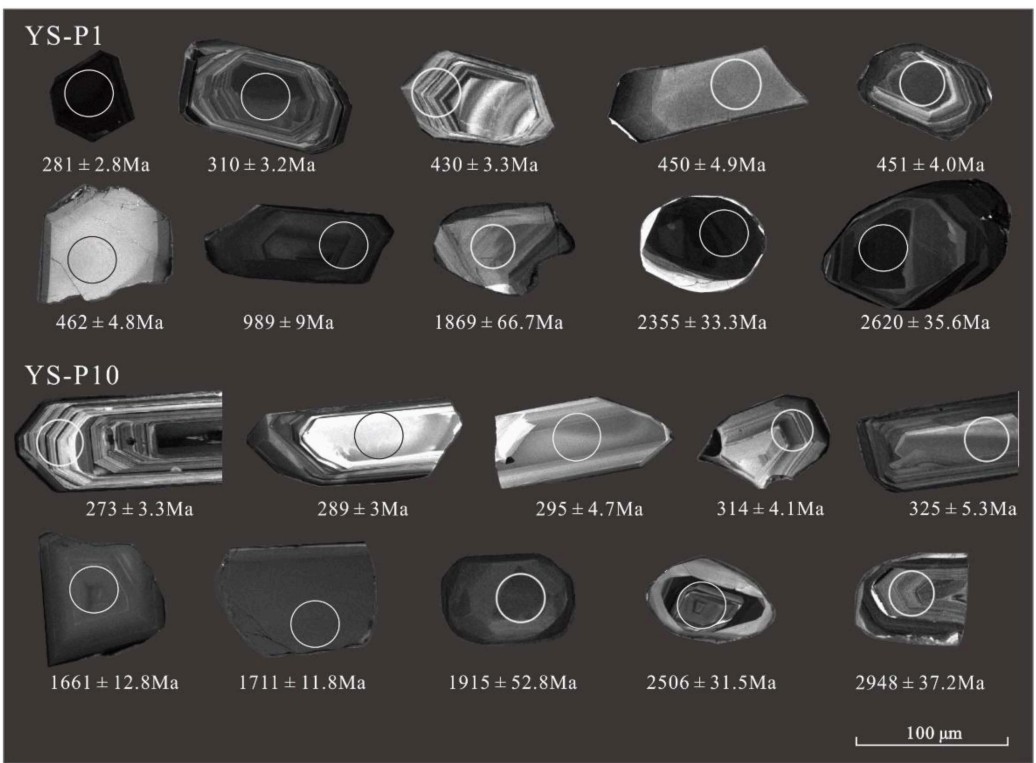

**Figure 3.** Representative CL images and U-Pb ages of detrital zircons from the two samples showing variations in size and morphology. The circles are analytical spots.

### 4.2. U-Pb Dating

A total of 177 zircons were selected from two samples for testing, and the influence of cracks and inclusions was avoided as far as possible in the selection of laser points. In total, 170 zircon U-Pb ages with a concordance of ≥90% were obtained from the two samples (Table S1). In the analysis of zircon U-Pb age data, $^{207}Pb/^{206}Pb$ age is generally used for ancient zircon (>1000 Ma) with more accumulated radiogenic components, and $^{206}Pb/^{238}U$ age is more appropriate for relatively young zircons (<1000 Ma) [50]. These data points are cast on zircon U-Pb concordia diagrams.

A total of 85 zircon U-Pb ages with a concordance of ≥90% were obtained from YS-P1 sandstone samples. The ages were grouped into three populations. The first group ranges from approximately 280 to 353 Ma (14 grains, 16.5%) and shows one peak at ca.

288 Ma. The second group ranges from approximately 400 to 1139 Ma (25 grains, 29.4%) and shows one major peak at ca. 448 Ma and one weak peak at 908 Ma. The zircons from the first and second groups have Th/U ratios ranging from approximately 0.160 to 1.296, with six spots <0.4, suggesting that most zircons have a typical magmatic origin. The third group ranges from approximately 1306 to 2620 Ma (46 grains, 54.1%) and shows two weak peaks at 1912 and 2420 Ma. The zircons from the third group have Th/U ratios ranging from approximately 0.097 to 2.746, with eleven spots <0.4 and one spot <0.1, suggesting that some zircons experienced metamorphic modification of different degrees in the later stage (Figure 4a).

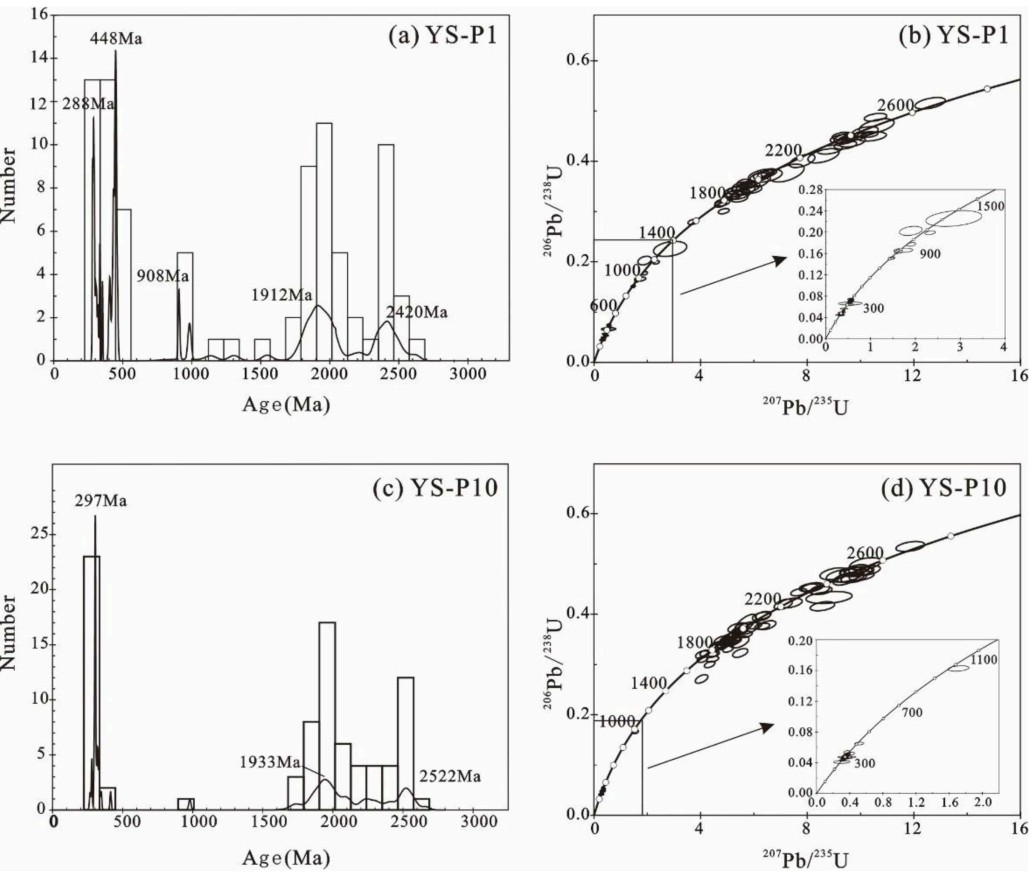

**Figure 4.** Results of detrital zircon U-Pb age histograms and concordia diagrams. (**a**) YS-1 histograms of U–Pb ages; (**b**) YS-1 concordia diagrams; (**c**) YS-10 histograms of U–Pb ages; (**d**) YS-10 concordia diagrams.

A total of 85 zircon U-Pb ages with a concordance of ≥90% were obtained from YS-P10 sandstone samples. The ages were grouped into two main populations. The first group ranges from approximately 259 to 339 Ma (24 grains, 28.2%) and shows one peak at ca. 297 Ma. The zircons have Th/U ratios ranging from approximately 0.576 to 1.136, suggesting that all zircons are of magmatic origin. The second group ranges from approximately 1700 to 2655 Ma (59 grains, 69.4%) and shows two weak peaks at ca. 1933 and 2522 Ma. The zircons have Th/U ratios ranging from approximately 0.094 to 8.428, with thirteen spots <0.4 and three spots <0.1, suggesting that some zircons experienced metamorphic modification of different degrees in the later stage (Figure 4c).

## 5. Discussion

### 5.1. Depositional Age

The time-equivalent strata of the inner North China Block are divided into the Xiashihezi and Shangshihezi formations, but the restriction of the Shihezi Formation in the

Luonan area made it difficult to compare our data with time-equivalent strata of the North China Block. In order to facilitate the comparison, we attempted to classify the attribution of the samples.

In general, the youngest concordant age of detrital zircon commonly reveals the maximum age of a sediment deposition, provided that the U-Pb system of the investigated zircon has not been disturbed during subsequent tectono-metamorphic or even hydrothermal events [51]. In order to constrain the maximum depositional age of the sedimentary succession, the minimum concordant U-Pb age (discordances < 5%) of magmatic zircon with clear oscillatory zones and Th/U ratios > 0.4 for each sample was selected. The youngest single zircon age from sample YS-P1 is 281 $\pm$ 3 Ma (Th/U = 0.74), close to the youngest age of the detrital zircon from the Xiashihezi Formation in the Qinshui area (276 $\pm$ 4 Ma) [17]. The youngest single zircon age from sample YS-P10 is 259 $\pm$ 5 Ma (Th/U = 0.75), also close to the youngest age of the detrital zircon from the Shangshihezi Formation in the Qinshui area (260 $\pm$ 4 Ma) [17]. Combined with the biostratigraphic age and sequence stratigraphic data [42,52], we suggest the sample YS-P1 belongs to the Xiashihezi Formation and sample YS-P10 is equivalent to the Shangshihezi Formation.

*5.2. Provenance Analysis*

Based on our U-Pb ages of detrital zircons ages, the two samples, YS-PS1 and YS-P10, can be divided into four age groups: the Late Paleozoic (353~259 Ma), Early Paleozoic (462~400 Ma), Mesoproterozoic–Neoproterozoic (1546~906 Ma) and Neoarchean–Paleoproterozoic (2655~1700 Ma) groups.

Late Paleozoic detrital zircons are widely distributed in the Permian–Triassic strata in the North China Block, and most authors suggest that these zircons originally came from the Inner Mongolia Palaeo-Uplift in the northern margin of North China Block [16,17,19]. The zircon U-Pb ages of Late Paleozoic magmatic rocks in the Inner Mongolia Palaeo-Uplift range from 390 to 237 Ma [53]; these zircons record tectonic uplift and volcanic magmatism under the influence of the subduction and closure of the paleo-Asian Ocean. The Inner Mongolia was uplifted and denuded at least 15 km from the Late Carboniferous to the Early Jurassic, providing a large amount of Late Paleozoic detrital zircons for the strata of the North China Block. The ages of ~290–300 Ma are mostly considered to be the typical peak of zircon ages [16,54,55], which is supported by the evidence of the $\varepsilon_{Hf}(t)$ value of a large number of Carboniferous–Permian I-type granites [56]. Paleogeographic study suggests that [42] the extension of the delta depositional system and the paleocurrent directions are all from north to south. This evidence strongly supports the concept that the Late Paleozoic zircons were sourced from the Inner Mongolia Palaeo-Uplift on the northern margin of the North China Block.

Early Paleozoic and Neoproterozoic zircons ages are the characteristic ages of the Qinling Orogenic Belt due to the intense tectonic-magmatism that occurred in these two periods. The Neoproterozoic tectonic-magmatism represented the collision between the North Qinling Belt and the North China Block during 1.0–0.8 Ga [57,58]. The Early Paleozoic tectonic-magmatism mainly occurred in the North Qinling Belt during 510–380 Ma, which was associated with the evolution of the Shangdan Ocean [1,26], and eventually caused the closure of Proto-Tethys Ocean in East Asia [59]. These two age groups are not developed in the inner and northern margin of North China Block, so the Neoproterozoic and Early Paleozoic detrital zircons can only be sourced from the Qinling Orogenic Belt.

Neoarchean–Paleoproterozoic detrital zircon ages are consistent with the basement of the North China Block. The North China Block is one of the oldest cratons in the world which experienced the rapid accretion of crustal growth at ~2.5 Ga and the amalgamation of the eastern and western blocks along the Trans-North China Orogen at ~1.85 Ga [60,61]. In particular, the tectonic magmatic event of ~2.5 Ga is considered to be the main time of the Precambrian crustal growth and reconstruction of the Paleo-North China Block [62–64], while ~1.85 Ga represents a unique prominent age peak of the Precambrian North China Block. It is considered to be related to the aggregation of the Columbia supercontinent [29,30,65].

To sum up, the zircon ages of sample YS-P1 can be divided into four age groups: Late Paleozoic (peak at 288 Ma), Early Paleozoic (peak at 448 Ma), Mesoproterozoic–Neoproterozoic (peak at 908 Ma) and Neoarchean–Paleoproterozoic (peak at 1912 Ma and 2420 Ma). Therefore, the provenance of sample YS-P1 was sourced from the Inner Mongolia Palaeo-Uplift, North Qinling Belt and the basement of North China Block. The zircon ages of samples YS-P10 can be divided into two age groups: Late Paleozoic (peak at 297 Ma) and Neoarchean–Paleoproterozoic (peak at 1933 Ma, 2522 Ma). Compared to sample YS-P1, Early Paleozoic and Meso-Neoproterozoic age groups are absent, suggesting the same sources as those from the base of the sequence apart from the North Qinling Belt.

### 5.3. Detrital Zircon Age Composition from the Late Paleozoic Strata

Comparing the detrital zircon U-Pb ages in this paper with the age composition of the Late Paleozoic strata in the southern North China Block and the Permian strata in the South Qinling area (Table 1, Figure 5) and analyzing the temporal and spatial distribution of detrital zircon U-Pb ages is helpful to understand the paleogeography of the southern North China Block and the Late Paleozoic tectonic evolution of the Qinling Orogenic Belt.

**Table 1.** Collection and summary of detrital zircon U-Pb ages of late Paleozoic strata in southern North China Block and South Qinling Belt.

| Stratigraphic Chronology | Sampling Location and Strata | Location (Lat/Long) | Lithology and Sample Number | U-Pb Age Composition (%)/Ma | | | Zircon Quantity/Pcs | Data Sources |
|---|---|---|---|---|---|---|---|---|
| Late Permian | Shiqianfeng Formation in Yiyang | / | Sandstone Y-4 | 357~242 (31%) | 1254~441 (9%) | 2733~1722 (60%) | 74 | Wang et al. [19] |
| | | | Sandstone Y-3 | 420~377 (16%) | 1433~423 (71%) | 2537~1587 (13%) | 74 | |
| | | | Sandstone Y-2 | 392~259 (32%) | 543~442 (4%) | 2631~1684 (64%) | 73 | |
| | Shiqianfeng Formation in Qinshui | 35°51′1.8″ N 112°22′2.4″ E | Fine-grained lithic quartz sandstone YC07 | 415~227 (12%) | 1037~456 (3%) | 2658~1402 (85%) | 103 | Zhu et al. [17] |
| | Permian in Zhen'an | / | Sandstone sxz-2 | 285~256 (3%) | 1359~448 (28%) | 2619~1554 (68%) | 88 | Cheng et al. [66] |
| | | | Sandstone sxz-1 | 393~256 (11%) | 1143~447 (11%) | 2666~1673 (79%) | 96 | |
| | | | Sandstone sx079 | 378~265 (4%) | 1424~438 (18%) | 2731~1717 (78%) | 83 | |
| Middle Permian | Shangshihezi Formation in Luonan | 34°01′48.00″ N 110°07′12.00″ E | Sandstone YS-P1 | 406~259 (29%) | / | 2655~1700 (71%) | 85 | This study |
| | Xiahihezi Formation in Luonan | 34°01′48.00″ N 110°07′12.00″ E | Sandstone YS-P10 | 413~280 (20%) | 1306~424 (27%) | 2620~1546 (53%) | 85 | |
| | Shangshihezi Formation in Yiyang | / | Sandstone Y-1 | 312~284 (12%) | / | 2561~1769 (88%) | 75 | Wang et al. [19] |
| | Xiashihezi Formation in Dengfeng | 112°58′56″ E 34°20′07″ N | Khaki-colored feldspathic quartz sandstone 13H7 | 336~276 (20%) | 1161~461 (5%) | 2529~1650 (75%) | 59 | Yang et al. [20] |
| | Shangshihezi Formation in Qinshui | 35°37′15.4″ N 112°23′23.7″ E | Coarse-grained lithic quartz graywacke YC04 | 415~274 (26%) | 1542~1258 (2%) | 2709~1611 (72%) | 97 | Zhu et al. [17] |
| | Xiahihezi Formation in Qinshui | 35°32′14.2″ N 112°24′34.7″ E | Coarse-grained lithic quartz sandstone YC03 | 401~260 (21%) | 1330~428 (2%) | 2807~1647 (77%) | 108 | |
| | Shanxi Formation in Qinshui | 35°28′1.2″ N 112°25′21.4″ E | Medium-grained graywacke YC02 | 396~279 (29%) | 451 | 2551~1691 (70%) | 101 | |
| Late Carboniferous–Early Permian | Taiyuan Formation in Qinshui | 35°23′6.7″ N 112°26′10.8″ E | Coarse-grained quartz sandstone YC01 | 410~406 (2%) | 1451~420 (69%) | 3359~1693 (29%) | 129 | |
| | Taiyuan Formation in Dengfeng | 34°33′36″ N 113°29′17″ E | Siltstone 13H1 | / | 1088~439 (90%) | 2507~1603 (10%) | 59 | Yang et al. [20] |

**Table 1.** *Cont.*

| Stratigraphic Chronology | Sampling Location and Strata | Location (Lat/Long) | Lithology and Sample Number | U-Pb Age Composition (%)/Ma | | | Zircon Quantity/Pcs | Data Sources |
|---|---|---|---|---|---|---|---|---|
| Late Carboniferous | Benxi Formation in Shagnwutou | / | SWT | / | 996~443 (65%) | 2615~1901 (35%) | 32 | Wang et al. [67] |
| | Benxi Formation in Gongyi | 34°34′37.2″ N 112°57′52.0″ E | Bean oolitic bau-xite 620-1 | 418~393 (6%) | 894~423 (86%) | 2714~1678 (9%) | 114 | Cao et al. [18] |
| | Benxi Formation in Hebi | / | Mudstone 180319-2 | 415~304 (8%) | 1144~425 (52%) | 2813~1610 (40%) | 78 | Sun [68] |

The detrital zircon U-Pb ages of the Late Carboniferous–Early Permian Benxi Formation and the Taiyuan Formation in the southern North China Block are mainly concentrated in the Neoproterozoic–Early Paleozoic, forming a main peak age of ~440 Ma. These detrital zircons are sourced from the Qinling Orogenic Belt, while the sediments from the Inner Mongolia Palaeo-Uplift can hardly be recorded in the southern North China Block. This feature suggests that even if long-term weathering and denudation occurred after Caledonian orogeny, the Qinling Orogenic Belt still had a fairly high paleotopography during the Late Carboniferous–Early Permian. The uplift of Inner Mongolia Palaeo-Uplift in the northern margin of the North China Block has continued consistently since the Early Carboniferous, but the paleotopographic features of "high in the south and low in the north" since the Early Paleozoic have not been changed.

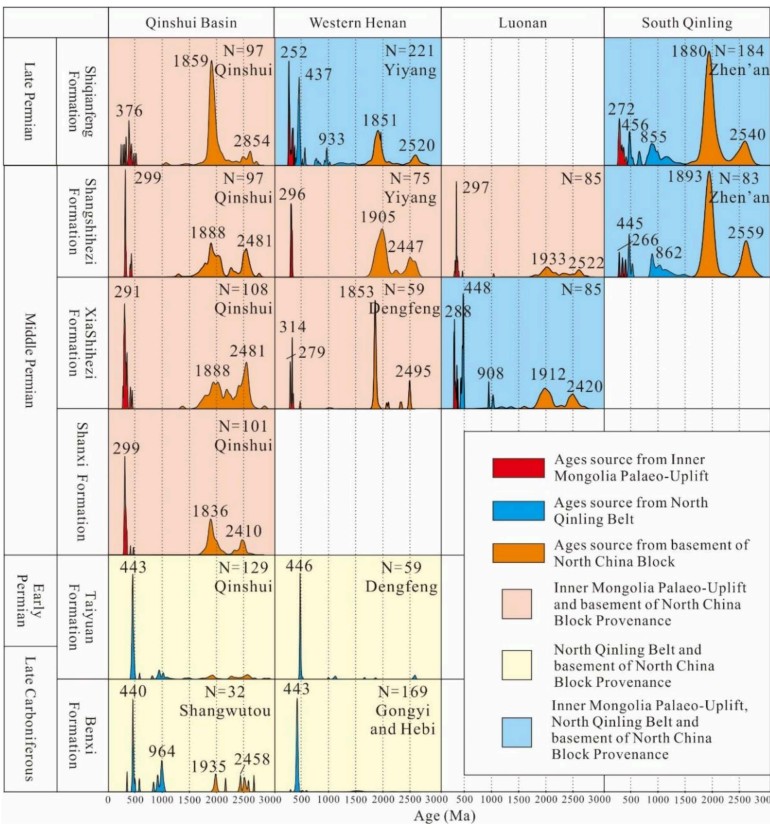

**Figure 5.** Comparison of detrital zircon U-Pb age spectrum obtained from south North China Block (data sources: Qinshui from Zhu et al. [17], Shangwutou from Wang et al. [67], Gongyi from Cao et al. [18], Hebi from Sun [68], Dengfeng: Taiyuan Formation from Yang et al. [20]; Xiashihezi Formation from Wang et al. [19], Luonan from this study, Zhen'an from Cheng et al. [66]).

The zircon U-Pb age compositions of the Middle Permian Shanxi Formation, Shangshihezi Formation and Xiashihezi Formation are relatively consistent, which are mainly aged from the Late Paleozoic and Neoarchean–Paleoproterozoic. These detrital zircons were mainly sourced from the Inner Mongolia Palaeo-Uplift on the northern margin of North China Block, and the Qinling Orogenic Belt could hardly provide sediments for these strata. However, according to the age composition of the Shihezi Formation in the Luonan area on the southern margin of North China Block, samples YS-P1 simultaneously recorded sources from the Inner Mongolia Palaeo-Uplift and Qinling Orogenic Belt, indicating that the depositional center was located in the Luonan area during this period. In contrast, sample YS-P10 only recorded sources from the Inner Mongolia Palaeo-Uplift. Moreover, detrital zircons from the Middle–Late Permian strata in the South Qinling area are composed of Late Paleozoic, Early Paleozoic, Neoproterozoic and Neoarchean–Mesoproterozoic grains [66] (Figure 5). This age composition indicated that the sediments should have been sourced from the Qinling Orogenic Belt, Inner Mongolia Palaeo-Uplift and North China Block. These provenance characteristics suggest that the Inner Mongolia uplift had been strongly uplifted and had the ability to provide sediment for the southern margin of the North China Block, even in the South Qinling area. Thus, the uplift of the Qinling Orogenic Belt was weakened, basically losing the ability to provide clastic matters to the Luonan area in the late Middle Permian. The North China Block has completed the tectonic transformation process from "high in the south and low in the north" in the early Late Paleozoic to "high in the north and low in the south" in the late Late Paleozoic.

*5.4. Tectonic Implications for Qinling Orogenic Belt*

The study of detrital zircon chronology from the Late Carboniferous to Early Permian in the southern North China Block shows that the Qinling Orogenic Belt was in a state of denudation during this period, and the lack of time-equivalent strata in the Luonan area also proves this view. This can be explained from two aspects: one is the continuation of the Early Paleozoic orogeny in the Qinling Orogenic Belt; the other is that the Qinling Orogenic Belt experienced orogenic uplift again in the Late Carboniferous.

The Early Paleozoic orogeny of the Qinling Orogenic Belt is related to the closure of the Shangdan Ocean. According to the age of mafic rocks and subduction-related rocks in the Shangdan ophiolite belt [2] and Cambrian–Ordovician radiolarian fossils in basalt interlayers in the Danfeng area [69], the formation time of the Shangdan Ocean basin is limited to the Early Cambrian [2] or Neoproterozoic [24]. Subsequently, the Shangdan Oceanic crust began to subduct under the North China Block since the Late Cambrian, resulting in the development of the south margin of the North China Block into an active continental margin in the Early Paleozoic and the formation of the Erlangping back-arc basin [1]. Based on the crystallization ages of subduction-related gabbro and granite intrusions in the North Qinling terrane, as well as the peak metamorphic ages of HP/UHP metamorphic rock belts on the south and north sides of the Qinling Group, most scholars agree that the Shangdan Ocean basin closed in the Early Devonian [1,24,26,70] or before the Devonian [71–74]. This is also supported by the geochemical characteristics of sedimentary records of the Devonian Liuling Group [72,73] and Wuguan Group [75] in the South Qinling Belt. With the closure of the Shangdan Ocean basin in the Devonian, the South Qinling terrane experienced continental subduction and collided with the North Qinling arc, which resulted in the orogenic uplift and basement exhumation of the North Qinling Belt in the Late Paleozoic [1]. The uplift lasted from the Late Carboniferous to Early Permian, which could be able to provide a large amount of sediments from the Qinling Orogenic Belt to the North China Block (Figure 6a).

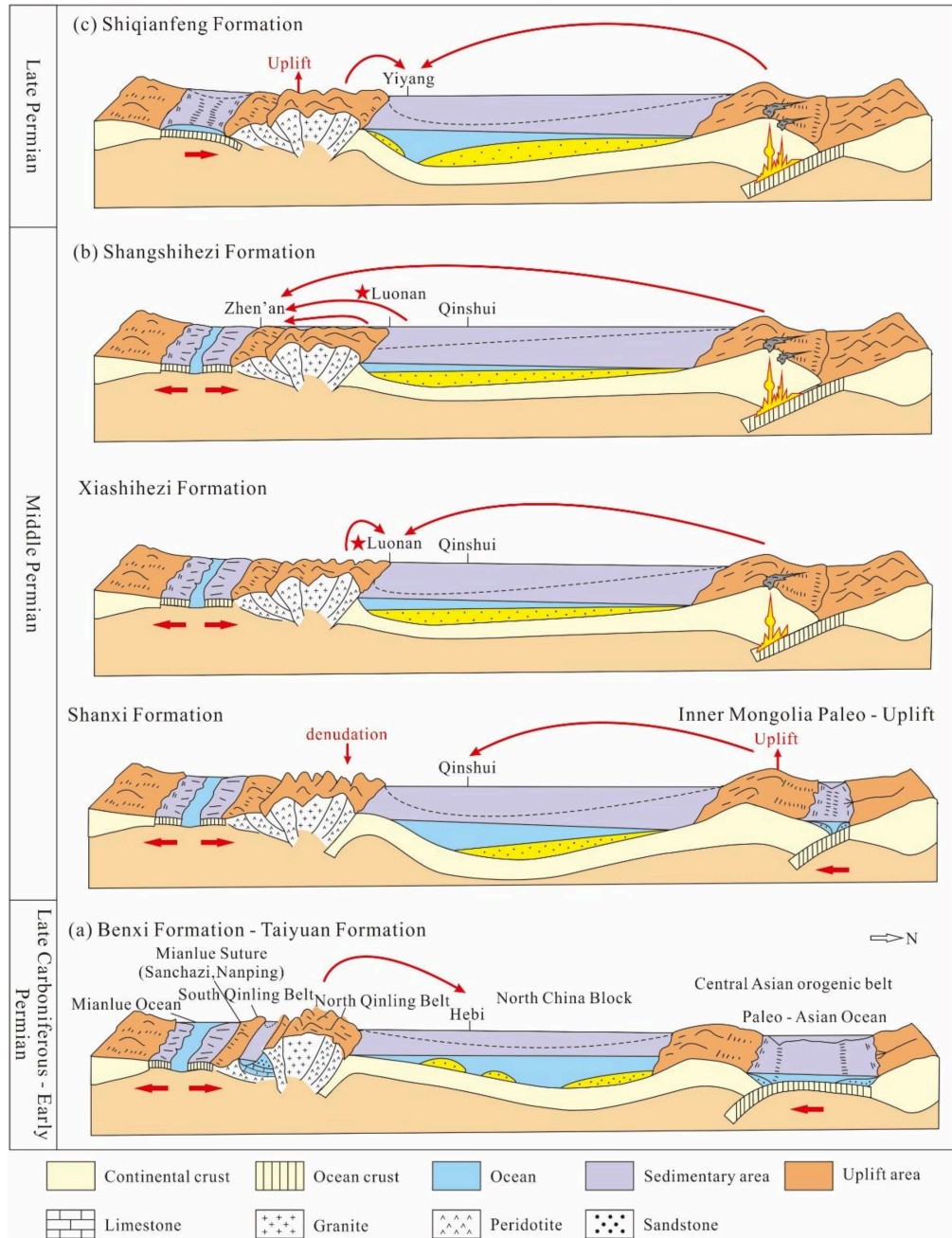

**Figure 6.** Late Carboniferous–Late Permian tectonic evolution in North China Block based on provenances. (**a**) The subduction stage of Paleo-Asian Ocean and the expansion stage of Mianlue Ocean; (**b**) the subduction-closure stage of Paleo-Asian Ocean; (**c**) initiation of the Mianlue Ocean subduction.

The Late Paleozoic tectonism of the Qinling Orogenic Belt is closely related to the evolution of the Mianlue Ocean basin. The Mianlue Ocean is the northern branch of the eastern Paleo-Tethys tectonic domain. Its occurrence, development and extinction have formed another important suture zone in the Qinling Orogenic Belt, namely the Mianlue Suture Zone. According to the analysis of petrology and geochemistry of ophiolites and related volcanic rocks in the Mianlue Suture Zone, the Mianlue Ocean basin evolved from the Silurian rift to a limited ocean basin and fully opened in the Middle–Late Devonian; then, the Qinling Orogenic Belt stepped into a new plate tectonics stage from the Late Paleozoic to Triassic [2,76]. One view holds that the subduction of the Mianlue Ocean basin began in the Late Carboniferous. The anorthosite and diabase from the Sanchazi

area and andesite at the Nanping area yielded U-Pb zircon ages of 300 ± 61 Ma [8], 295–264 Ma [13] and 246 ± 3 Ma [14], respectively; these ages suggest that the Mianlue Ocean subducted during 300–250 Ma. The presence of the ophiolite, subduction-related magmatic associations and metamorphic records strongly argue an East-Pacific-type active continental margin. In this way, the Qinling Orogenic Belt enters the background of subduction orogeny again, which will inevitably lead to the uplift of the North Qinling Belt to a certain extent and provide a large amount of sediments for the Late Carboniferous–Permian strata of the North China Block. However, a large area of sedimentary strata was distributed in the Qinling Orogenic Belt in the Permian [77]. In addition, there are sediments sourced from the Inner Mongolia Palaeo-Uplift in the northern margin of North China Block, but there is lack of sediments sourced from the Qinling Orogenic Belt in the North China Block (Figure 6b). Sedimentary records in the southern margin of the North China Block indicate that the North Qinling Belt should have been relatively tectonically quiet in the Middle Permian, without significant orogeny. From this aspect, the South Qinling lithosphere was mainly in an extended state, indicating that the expansion of the Mianlue Ocean Basin may still continue. Thus, the North Qinling Belt suffered subsidence and can hardly provide sediments for the southern North China Block. In addition, isotopic age and paleontological evidence suggest that the opening and development of the Mianlue Ocean basin should have been mainly in the Middle–Late Devonian to Middle Permian [8,78,79]. The above evidence is more supportive of the concept that the Mianlue Ocean basin was still dominated by extension during the Middle Permian. Thus, we support that the Late Carboniferous uplift of the Qinling Orogenic Belt is the continuation of the Early Paleozoic orogeny.

The possible subduction time of the Mianlue Ocean may be in the Late Permian, and the initial collision occurred in the Early Triassic. The evidence is as follows: (1) The paleocurrent directions in the southern North China Block have changed from north to south in the early period to south to north in the late period, indicating that the provenance area has changed [80,81]. (2) An extensive northward progradation delta system formed in the southern North China Block in the Late Permian, and the clastic composition determined that the southern North China Block and the North Qinling Belt may have initially uplifted in the Late Permian (253 Ma) [82]. (3) The geochemical characteristics of clastic rocks indicated that tectonic transformation occurred in the Late Permian Shiqianfeng Formation [83]. (4) The finer clastic zircon chronology evidence indicated that the provenance from the Qinling Orogenic Belt and the southern margin of North China Block began to appear during the Late Permian [19,20]. (5) According to Wang's [84] study on the Shiqianfeng Formation in western Henan, the Pingdingshan sandstone member is conformably in contact with the underlying Shangshihezi Formation in most areas, but there are still some areas showing stratigraphic discontinuities. Liu [85] found that angular unconformity existed between the Pingdingshan sandstone member and the Shangshihezi Formation in the Dengfeng area. Wu [86] also argued that there is unconformity from the Late Cisuralian to Guadalupian (ca. 280–260 Ma) between the Shangshihezi and Sunjiagou Formations.

Through these data of the Mianlue Ocean in the Paleo-Tethys tectonic domain, which began to transform from expansion and extension to subduction and compression in the Late Permian (~ca. 260 Ma) (Figure 6c), causing the North Qinling Belt and the southern margin of North China Block to be re-uplifted. The initial uplift may be consistent with the sedimentary transition between the Xiashihezi Formation and the Pingdingshan sandstone member. At the beginning of the Early Triassic, abundant detritus from the Qinling Orogenic Belt appeared in the southern North China Block [12,87], which is direct evidence of the collision between the South China Block and the North China Block.

## 6. Conclusions

(1) The detrital zircon U-Pb age result shows that sample YS-P1 from the bottom of the Permian strata in the Luonan area shows two major peaks at 288 Ma and 448 Ma and three weak peaks at 908 Ma, 1912 Ma and 2420 Ma, which indicates that sediments were sourced from the Inner Mongolia Palaeo-Uplift, North Qinling Belt and the basement of the North China Block. Sample YS-P10 from the top of Permian strata shows one major peak at 297 Ma and two weak peaks at 1933 and 2522 Ma, which indicated that the provenance was the Inner Mongolia Palaeo-Uplift and the basement of North China Block.

(2) The Late Carboniferous uplift of the Qinling Orogenic Belt is the continuation of the Early Paleozoic orogeny, which resulted in abundant detritus transported from the North Qinling Belt to the North China Block. The Mianlue Ocean continued to expand until the Middle Permian, causing the North Qinling Belt to be in an extensional tectonic setting, which basically lost the ability to provide sediments to the southern North China Block in the late Middle Permian. We suggest that the subduction might have occurred in the sedimentary transition between the Shangshihezi Formation and the Shiqianfeng Formation (ca ~260 Ma), which led to the uplift of the North Qinling Belt and the southern margin of North China Block again.

**Supplementary Materials:** The following are available online at https://www.mdpi.com/article/10.3390/min12070864/s1, Table S1: LA-ICP-MS detrital zircon U-Pb ages from the Luonan area.

**Author Contributions:** Conceptualization T.F. and Y.W.; formal analysis, Y.W.; investigation, H.S. and W.Y.; project administration, Y.W.; writing—original draft, T.F.; writing—review and editing, Y.W. funding acquisition, Y.W. All authors have read and agreed to the published version of the manuscript.

**Funding:** This research was funded by the Outstanding Youth Science Foundation of Henan Province, grant number 222300420046.

**Data Availability Statement:** The data presented in this study are available in supplementary material.

**Conflicts of Interest:** The authors declare no conflict of interest.

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
