# Peer review of "Late Paleozoic Tectonic Evolution of the Qinling Orogenic Belt: Constraints of Detrital Zircon U-Pb Ages from the Southern Margin of North China Block"

_minerals, doi:10.3390/min12070864_

Round 1
Reviewer 1 Report
This manuscript by Yang et al. presents detrital zircon U-Pb ages from the southern margin of North China Block to reconstruct the provenance and tectonic evolution of Qinling Orogenic Belt. Overall, the MS is well-written, and presents an intriguing data set to discuss the provenance changes and tectonic activities. The paper subject is undoubtedly suitable for Minerals, and I recommend the following revisions to the MS:
P1 L36-37: “South China Block, the Qinling microblock, and the North China Block”, consider adding a simplified tectonic map figure showing the major units of the area, the current Fig 1 is pretty detailed and not easy to dive in directly at the beginning. A simplified map showing these big tectonic blocks/units would be intuitive to set up a big picture and help readers understand the evolution.
P2 L45: “Sanchazi area, Nanping area”, please show these on the map figure, we don’t know where they are.
P2 L70 and L72: “Lingbao-Lushan-Wuyang” “Mianlue-Bashan-Xiangguang”, please show these on the map figure, we don’t know where they are.
P7 L233 and L236: “maximum age”, it’s not appropriate to use maximum, please change to youngest/oldest when describing ages.
P8 L248-249: add the numerical ages to the four groups
P8 L246 5.2 Provenance Analysis: it will be more intuitive and clear to use KDE plot to compare your analyzed U-Pb ages with all the potential source area ages if available.
Figure 1: Outline the area of figure b in figure a, otherwise we don’t know their relationship.
Figures 5 and 6: add the analyzed ages in this study at the top of both figures for comparison. What are the colored area?
Figure 7: add the figure legend
Author Response
First of all, thank you very much for you comments,your guidance is of great significance. My modifications are as follows:
Point1:P1 L36-37: “South China Block, the Qinling microblock, and the North China Block”, consider adding a simplified tectonic map figure showing the major units of the area, the current Fig 1 is pretty detailed and not easy to dive in directly at the beginning. A simplified map showing these big tectonic blocks/units would be intuitive to set up a big picture and help readers understand the evolution.
P2 L45: “Sanchazi area, Nanping area”, please show these on the map figure, we don’t know where they are.
P2 L70 and L72: “Lingbao-Lushan-Wuyang” “Mianlue-Bashan-Xiangguang”, please show these on the map figure, we don’t know where they are.
Response 1: Since the main research objects are the North Qinling orogenic belt and the North China plate, and the tectonic units of the Qinling Mountains and the North China plate are introduced in the previous introduction, please allow me to keep this picture for the sake of corresponding to the previous article. I further modified the picture by adding sanchezi, Nanping area and two fault zones in the notes below the picture.
Point2:P7 L233 and L236: “maximum age”, it’s not appropriate to use maximum, please change to youngest/oldest when describing ages.
P8 L248-249: add the numerical ages to the four groups
Response 2: Modified according to your opinion.
Poing3: P8 L246 5.2 Provenance Analysis: it will be more intuitive and clear to use KDE plot to compare your analyzed U-Pb ages with all the potential source area ages if available
Response 3: I think the original plot may be able to compare the age peak more intuitively than the KDE plot. In addition, I further added the zircon age plot of the source area. Please allow me to keep the original figure.
Point 4: Figure 1: Outline the area of figure b in figure a, otherwise we don’t know their relationship.
Figures 5 and 6: add the analyzed ages in this study at the top of both figures for comparison. What are the colored area?
Figure 7: add the figure legend
Response 4: Modified according to your opinion.
Reviewer 2 Report
General Comments
The manuscript is generally well constructed and presents interesting new data. Most of the figures are well-drafted and illustrate the concepts needed for the arguments presented. However, a few modifications in the text, figures, and captions could make it easier to read and to follow. Also, some additions will help to trust the geochronology data presented.
English is my second language, so I am not the best person to judge the language level of this paper. However, few grammatical turns seem to me to be flawed and required a special attention in order to try to understand what the authors meant. In my line-by-line comments, I have underlined or corrected some of the passages that gave me the most difficulties, but I suggest that the final version be proofread by an English-speaking person.
Please provide GPS coordinates for all the samples in this study, preferably in Latitude and Longitude in WGS84 datum or UTM (without forgetting to specify the zone). Be sure to include a statement that tells the reader the datum. I cannot find sample coordinates or location descriptions in the manuscript or supplementary material. The value of the paper is greatly reduced if readers cannot tell precisely where your samples came from. A brief table with sample number, description of the sample, outcrop and location such as road cut or natural outcrop, and GPS coordinates is invaluable and most certainly should be included. This information could be entered into the supplementary data table as a header next to the sample number, and thus would not require a separate table. You could also add some of these information s in your Table 1.
Regional Geology
There is a lot of information here, and a nice effort for clarity. However, I think that the Shihezi Formation should be introduced much earlier than it is. For a long moment, I just wondered where your Permian strata of the Luonan area belonged to.
From the caption fig .2 (page 5, almost half of the manuscript) I understand that the Permian strata belong to the Shihezi Formation but you didn't mention it before, or it is not clear. I suggest mentioning the Shihezi Formation much earlier, in the Abstract and the Introduction and the Context. It might help to situate it compared to the overall Qinling Orogenic Belt too. It is very important to highlight these sediments, their position and context of deposition compared to the other geological objects around, as they are hosting your analyses. And for now it is not entirely clear.
Also please mention somewhere in the abstract and very early on in the introduction that the metamorphic grade is very low, and so detrital muscovite ages will record crystallization and hence source information.
General analytical procedures
The Zircon U-Pb LA-ICP-MS Geochronology method should be improved to highlight the good quality of the data. In the actual state, data seem good, mostly concordant, and precise. However, to judge the accuracy, the secondary standards data need to be reported in the supplementary material. Did you use a second independent reference material to make sure of the accuracy of your analysis? It should be used as good practice and mentioned clearly in the Analytical Method.
Also, please use the primary reference for the 91500 age standard (Wiedenbeck et al., 1995).
Wiedenbeck, M.A.P.C., Alle, P., Corfu, F.Y., Griffin, W.L., Meier, M., Oberli, F.V., Quadt, A.V., Roddick, J.C. and Spiegel, W., 1995. Three natural zircon standards for U‐Th‐Pb, Lu‐Hf, trace element and REE analyses. Geostandards newsletter, 19(1), pp.1-23.
Result
Only two detrital samples have been analysed, the manuscript would been stronger at least with a third one to see if the transition is progressive or drastic, and that there is a reproducibility of the results, making sure that the differences are not just an effect of topography, river pathways, channels, etc.. Is there a reason why only these two sandstone units have been sampled? Why the sandstones units in between have not been sampled / or analysed? If there was a difficult access to these units, or other difficulties, it would be worth mentioning.
I don't understand the grouping of the zircon population for the various samples. In my opinion, the two samples have a probability density plot that show two groups. For the first one the first group includes the two peaks at 288 and 448 Ma, the second one with Meso and Paleoproterozoic ages. For the second sample I agree with your two groups. But so, I wonder why you do three groups in the first sample? If you keep this idea of three groups, please explain briefly why line 205.
Discussion
Because of the language, the first paragraph gave me some difficulties and required several rereading’s in order to try to understand what the authors meant. I tried to suggest some corrections in my line-by-line comments in the PDF, however, sentence constructions and grammatical turns seem to me to be flawed and should be tidy up. Also, I think part of the problem is that you introduce a new problem here, that was not raised at all from the beginning: To which Formation the Permian Strata of Luonan are belonging? This problematic should be raised in introduction and context. Maybe that’s why there is a bit of confusion in the introduction and the context of what are the Permian strata.
Youngest detrital age calculations
You need to be careful when interpreting maximum depositional ages. There is a big discussion about what to use as the maximum depositional age. Using The youngest concordant age of detrital zircon is not the most conservative approach, and I would say not the most common approach for maximum depositional ages as suggested line 233. Most people will use a weighted mean of the youngest population or the youngest few grains, which is much more conservative, as we do not want erroneously young depositional ages entering the scientific literature.
However, this approach seems to fit with the regional geology and the correlated strata. And I think it would be a better reason to mention when you justify why you choose to use this method for interpreting your maximum depositional age instead of the more common weighted mean of the youngest population (line 233).
Line 244. The Xiashihezi and Shangshihezi Formations, to which are correlated the Permian strata of this study are not described or defined before the discussion. I wonder if it would not help the reader to link everything if they were introduced in the geological context or introduction, along with the question of the belonging of these Permian strata in Luonan area.
Provenance Analysis
I thought a reasonable job had been done here, however, this section needs a bit of tidying up, the first paragraph has quite a lot of repetitions. Also, the mention of ages that are not present in the samples without clear reason got me confused. Please, re-carefully read this part, cutting the repetitions and removing any not essential information.
In my line-by-line comment in the PDF I suggest some ways to make it clearer and/or shorter.
Detrital Zircon Age Composition from the Late Paleozoic strata
In the table 1. Please add your samples (data source: This study).
Also, I would strongly suggest adding a column for the location (UTM or Lat/Long)
Tectonic Implications for Qinling Orogenic Belt
I found it a bit difficult to read this last part of the discussion. Needing to go back and forth in between the data, and the sketches, not always finding the information I was looking for at the first look. I suggest to really be careful in the order of presenting the data. Use your figure 7 to guide your text, describe it from a) to e), one step at a time, always referencing the figure at each step. Also, in the figure you could give some names at each stage in each of your little sketch, either on the figure or in the caption. For example: e) Initiation of the Mianlue Ocean Subduction.
I guess I had some trouble to locate exactly your samples from your study, and to do the link between the data and the sketch because all sources mentioned before were not highlighted on the figure. For example, the detrital zircon distribution in the samples from Zhen'an show more provenances than the Inner Mongolia Paleo-Uplift. Line 329, you say "This age composition indicated that the sediments should source from Qinling Orogenic Belt, Inner Mongolia Palaeo-Uplift and North China Block ", however you show only one arrow from the Mongolia Paleo-Uplift. You should represent all of them on your diagram.
Doing so will also help you to emphasis the weakening of the local sources compared to the Inner Mongolia Paleo-Uplift, with drawing provenances with big and small arrows , or double and simple arrows depending on the importance of the sources. Don’t hesitate to write UPLIFT, or DENUDATION, or any process you mention in the text on your figure, to help, once again the link between the two.
Figures
As said earlier, most of the figures are clear and illustrate the concepts needed for the arguments presented. However, I suggest below few modifications in the figures and captions that could make it even easier to read and to follow.
Also, be careful to have the figure appearing in the text in a sequential order. Right now, fig. 7b appears before 7a for example. It needs to be corrected and fixed for all figures.
Figure 1: some grey shades could highlight the sediments you are studying
The square of the location of the study area is very difficult to see. A red square named “study area” could be the way to go.
Separate the explanation in three columns: Stratigraphy; Structural symbols; Stratigraphic column.
Figure 2: Photo 1 decide if you want to highlight the Formation or Shihezi Formation or both and make it clear in the text.
Table 1: Add your samples
Fig. 5: I would suggest identifying your color bands by writing “Late Paleozoic”, “Neoproterozic to early Paleozoic”, etc on the top, or on the upper part of the figure - it will link them better with the text (and you can refer to them in the text too).
Fig. 6 : a, b, a needs to be fixed into a, b, c. Also, in caption you need to describe a, b, and c.
Fig. 7 : In caption, describe a, b, c, d. Also you can have a little title for each of them on the figure.
On your diagrams, locate your samples for the reader to be easier to link your diagram with the data.
It would be nice to locate the Permian andesites, diabase and anorthosite on your fig.7 if possible, and in the text discuss an alternative way to produce them than subduction.
Add the name of processes for each step UPLIFT, or DENUDATION. Be careful to show all the provenances
See the PDF for complementary and additional line by line comments.

Author Response
Thank you for your serious, detailed and instructive comments and suggestions. My modifications are as follows:
Point 1: Please provide GPS coordinates for all the samples in this study, preferably in Latitude and Longitude in WGS84 datum or UTM (without forgetting to specify the zone). Be sure to include a statement that tells the reader the datum. I cannot find sample coordinates or location descriptions in the manuscript or supplementary material. The value of the paper is greatly reduced if readers cannot tell precisely where your samples came from. A brief table with sample number, description of the sample, outcrop and location such as road cut or natural outcrop, and GPS coordinates is invaluable and most certainly should be included. This information could be entered into the supplementary data table as a header next to the sample number, and thus would not require a separate table. You could also add some of these information s in your Table 1.
Response 1: Modified according to your opinion.
Point 2:
Regional Geology
There is a lot of information here, and a nice effort for clarity. However, I think that the Shihezi Formation should be introduced much earlier than it is. For a long moment, I just wondered where your Permian strata of the Luonan area belonged to.
From the caption fig .2 (page 5, almost half of the manuscript) I understand that the Permian strata belong to the Shihezi Formation but you didn't mention it before, or it is not clear. I suggest mentioning the Shihezi Formation much earlier, in the Abstract and the Introduction and the Context. It might help to situate it compared to the overall Qinling Orogenic Belt too. It is very important to highlight these sediments, their position and context of deposition compared to the other geological objects around, as they are hosting your analyses. And for now it is not entirely clear.
Also please mention somewhere in the abstract and very early on in the introduction that the metamorphic grade is very low, and so detrital muscovite ages will record crystallization and hence source information.
Response 2: I introduced the problem that the Permian strata in Luonan area belong to Shihezi Formation earlier in this paper, so as to facilitate a better explanation below. At the same time, some information about sedimentary strata, geographical location and sedimentary environment is supplemented.
I have supplemented your suggestions in the abstract and introduction.
Point 3:
General analytical procedures
The Zircon U-Pb LA-ICP-MS Geochronology method should be improved to highlight the good quality of the data. In the actual state, data seem good, mostly concordant, and precise. However, to judge the accuracy, the secondary standards data need to be reported in the supplementary material. Did you use a second independent reference material to make sure of the accuracy of your analysis? It should be used as good practice and mentioned clearly in the Analytical Method.
Also, please use the primary reference for the 91500 age standard (Wiedenbeck et al., 1995).
Wiedenbeck, M.A.P.C., Alle, P., Corfu, F.Y., Griffin, W.L., Meier, M., Oberli, F.V., Quadt, A.V., Roddick, J.C. and Spiegel, W., 1995. Three natural zircon standards for U‐Th‐Pb, Lu‐Hf, trace element and REE analyses. Geostandards newsletter, 19(1), pp.1-23.
Response 3: Added a second independent reference material and use the primary reference for the 91500 age standard.
Point 4:
Result
Only two detrital samples have been analysed, the manuscript would been stronger at least with a third one to see if the transition is progressive or drastic, and that there is a reproducibility of the results, making sure that the differences are not just an effect of topography, river pathways, channels, etc.. Is there a reason why only these two sandstone units have been sampled? Why the sandstones units in between have not been sampled / or analysed? If there was a difficult access to these units, or other difficulties, it would be worth mentioning.
I don't understand the grouping of the zircon population for the various samples. In my opinion, the two samples have a probability density plot that show two groups. For the first one the first group includes the two peaks at 288 and 448 Ma, the second one with Meso and Paleoproterozoic ages. For the second sample I agree with your two groups. But so, I wonder why you do three groups in the first sample? If you keep this idea of three groups, please explain briefly why line 205.
Response 4: As mentioned above, the stratum exposed in the study area is thin, with a total length of less than 250m, and there is no landmark horizon. Before the test results are obtained, it is impossible to know whether the study stratum belongs to the upper Shihezi Formation / Lower Shihezi Formation. Therefore, we choose to test the top and bottom samples in order to obtain two different test results as far as possible under limited conditions, so as to try to further divide the stratum of the sample according to the test results. If the strata are clearly divided in the area, it is undoubtedly more conducive to interpretation to increase the number of test samples.
Please allow me to keep this idea of three groups. I understand your view on dividing age into two groups. The difference is that I divide the Paleozoic age into two groups: Early Paleozoic and late Paleozoic. This is convenient for the following discussion on the source areas. Because the early Paleozoic age corresponds to the North Qinling provenance, and the late Paleozoic age corresponds to the Inner Mongolia uplift provenance. If the ages of the two samples are broth divided into two groups here, it might not be clear enough when discussing the source area below.
Point 5: Discussion
Because of the language, the first paragraph gave me some difficulties and required several rereading’s in order to try to understand what the authors meant. I tried to suggest some corrections in my line-by-line comments in the PDF, however, sentence constructions and grammatical turns seem to me to be flawed and should be tidy up. Also, I think part of the problem is that you introduce a new problem here, that was not raised at all from the beginning: To which Formation the Permian Strata of Luonan are belonging? This problematic should be raised in introduction and context. Maybe that’s why there is a bit of confusion in the introduction and the context of what are the Permian strata.
Response 5: The problem of Shihezi Formation has been modified previously.
When using zircon age to explain the oldest sedimentary age, I must admit that the method used in this paper has certain risks. As you said, this approach sees to fit with the regional geography and the correlated strata. Our original intention is to compare the strata in Luonan area with those in North China, especially in Qinshui area. Considering that the youngest zircon age is used in Qinshui area to limit the oldest sedimentary age, for more accurate stratigraphic correlation, this paper also uses this method instead of the weighted average age method.
Point 6:
Provenance Analysis
I thought a reasonable job had been done here, however, this section needs a bit of tidying up, the first paragraph has quite a lot of repetitions. Also, the mention of ages that are not present in the samples without clear reason got me confused. Please, re-carefully read this part, cutting the repetitions and removing any not essential information.
In my line-by-line comment in the PDF I suggest some ways to make it clearer and/or shorter.
Detrital Zircon Age Composition from the Late Paleozoic strata
In the table 1. Please add your samples (data source: This study).
Also, I would strongly suggest adding a column for the location (UTM or Lat/Long)
Response 6: Modified according to your opinion.
Point 7:
Tectonic Implications for Qinling Orogenic Belt
I found it a bit difficult to read this last part of the discussion. Needing to go back and forth in between the data, and the sketches, not always finding the information I was looking for at the first look. I suggest to really be careful in the order of presenting the data. Use your figure 7 to guide your text, describe it from a) to e), one step at a time, always referencing the figure at each step. Also, in the figure you could give some names at each stage in each of your little sketch, either on the figure or in the caption. For example: e) Initiation of the Mianlue Ocean Subduction.
I guess I had some trouble to locate exactly your samples from your study, and to do the link between the data and the sketch because all sources mentioned before were not highlighted on the figure. For example, the detrital zircon distribution in the samples from Zhen'an show more provenances than the Inner Mongolia Paleo-Uplift. Line 329, you say "This age composition indicated that the sediments should source from Qinling Orogenic Belt, Inner Mongolia Palaeo-Uplift and North China Block ", however you show only one arrow from the Mongolia Paleo-Uplift. You should represent all of them on your diagram.
Doing so will also help you to emphasis the weakening of the local sources compared to the Inner Mongolia Paleo-Uplift, with drawing provenances with big and small arrows , or double and simple arrows depending on the importance of the sources. Don’t hesitate to write UPLIFT, or DENUDATION, or any process you mention in the text on your figure, to help, once again the link between the two.
Response 7: According to your opinion, I split and modified Figure 7.
Point 8:
Figure 1: some grey shades could highlight the sediments you are studying
The square of the location of the study area is very difficult to see. A red square named “study area” could be the way to go.
Separate the explanation in three columns: Stratigraphy; Structural symbols; Stratigraphic column.
Figure 2: Photo 1 decide if you want to highlight the Formation or Shihezi Formation or both and make it clear in the text.
Table 1: Add your samples
Fig. 5: I would suggest identifying your color bands by writing “Late Paleozoic”, “Neoproterozic to early Paleozoic”, etc on the top, or on the upper part of the figure - it will link them better with the text (and you can refer to them in the text too).
Fig. 6 : a, b, a needs to be fixed into a, b, c. Also, in caption you need to describe a, b, and c.
Fig. 7 : In caption, describe a, b, c, d. Also you can have a little title for each of them on the figure.
On your diagrams, locate your samples for the reader to be easier to link your diagram with the data.
It would be nice to locate the Permian andesites, diabase and anorthosite on your fig.7 if possible, and in the text discuss an alternative way to produce them than subduction.
Add the name of processes for each step UPLIFT, or DENUDATION. Be careful to show all the provenances
Response 8: According to your opinion, I modified all the Figure.
As for andesites and anorthosite, what I want to say is that I do not think it can be formed without subduction. The problem pointed out in this paper is that its age error range is wide (300 ± 61Ma), and the error range may include the Late Permian. Or the age span is large (295-264Ma), which can not accurately explain the subduction time limit of Mianlue ocean.
Round 2
Reviewer 2 Report
I think the authors have made good corrections, which have helped the manuscript to be clearer.
I have a few more details to pay attention to: I would suggest to mention in the caption of figure 1 that the grey area is the study area.
line 477: might and not minght.
Also, I think the figure 5 is much clearer than it was.
Author Response
Thank you very much for your affirmation and suggestions.
Figure 1 has been modified according to your suggestions.